# Effects of Aging Process on the Damping Performance of ZK60 Magnesium Alloys Prepared by Large Strain Rolling

**DOI:** 10.3390/ma13235574

**Published:** 2020-12-07

**Authors:** Xuhui Feng, Youping Sun, Siyu Wan, Gang Chen, Jiangmei He

**Affiliations:** 1School of Mechanical and Transportation Engineering, Guangxi University of Science and Technology, Liuzhou 545006, China; fengxuhui1995@gmail.com (X.F.); wansiyu1996@gmail.com (S.W.); jiangmeihe645@gmail.com (J.H.); 2Guangxi Key Laboratory of Automobile Components and Vehicle Technology, Guangxi University of Science and Technology, Liuzhou 545000, China; 3School of Materials Science and Engineering, Hunan University, Changsha 410082, China; chengang811@163.com

**Keywords:** ZK60 magnesium alloy, aging treatment, second phase, damping performance

## Abstract

In this study, the effects of an aging treatment (T5) and a solution + aging treatment (T6) on the microstructure and damping properties of a ZK60 magnesium alloy prepared by large strain rolling (LSR) were studied by an optical microscope (OM), scanning electron microscopy (SEM), X-ray diffraction (XRD), and dynamic thermomechanical analysis (DMA). The results showed that both the T5 and T6 processes had a great impact on the microstructure and damping properties of the ZK60 magnesium alloy. With the increase in aging time, the grain size was basically unchanged, and the amount of the second phase increased, resulting in a gradual decrease in the damping performance. However, compared with the damping performance of the un-aged ZK60 magnesium alloy, the damping performance of the 4 h-aged ZK60 magnesium alloy was enhanced. At the same aging time, the increase in the aging temperature promoted the precipitation of the second phase, thereby reducing the damping performance of the ZK60 magnesium alloy. It was found that the T6-treated ZK60 magnesium alloy had a larger grain size, which led to a better damping performance; in addition, the T6-treated ZK60 magnesium alloy exhibited a damping plateau, which was determined by the distribution and amount of the second phase.

## 1. Introduction

With the continuous development of modern industries, vibration and noise have become one of the three major public hazards. Mechanical vibration can cause noise, affect the accuracy of machines, resulting in breakage, and increase the loss of energy and raw materials. Many studies have shown that approximately two-thirds of industrial failures are caused by vibration and noise [1,2,3]. Therefore, the use of damping materials is extremely important for reducing the harm of vibration and noise. Currently, damping materials mainly include rubber, plastic damping plates, damping composite materials, and high damping alloys. As one of the best metallic damping materials, magnesium and magnesium alloys have the advantages of high specific strength, high specific stiffness, low density, and good damping performance. They are called “21st century green engineering materials” [4] and are widely used in aerospace, automotive, military, and other industries [5]. Therefore, the study of high damping magnesium alloys cannot only reduce the environmental pollution caused by vibration but also improve the service life of machines [6].

The ZK60 magnesium alloy is a widely used wrought magnesium alloy and has a strength close to that of the 7075 aluminum alloy, thereby having broad application prospects [7]. The damping mechanism of magnesium alloys is dislocation damping. To obtain high damping performances, dislocations must move as much as possible. However, to obtain good mechanical properties, the movement of dislocations must be limited. Therefore, dislocation-type damping materials generally exhibit a contradiction between damping properties and mechanical properties, which greatly limits their scope of use [8]. The common methods used to enhance mechanical properties include plastic deformation and aging treatment, of which the effects on the damping performance have been studied by many scholars. Lin et al. [9,10] found that after plastic deformation, the dislocation density in the alloy increased sharply, and many dislocation tangles formed. The dislocation tangles served as strong pinning points to hinder dislocation movement, thereby reducing the damping performance. The increase in the deformation extent promoted the recrystallization process. Thus, the number of grain boundaries increased, and more subcrystalline structures were obtained, thereby reducing the mobility of dislocations. Therefore, the damping performance of magnesium alloys significantly decreased as the deformation extent increased. Li et al. [11,12,13,14] found that the solution treatment caused the precipitates to dissolve into the matrix and reduced the number of second phase particles. At the same time, grains grew, and the number of dislocation tangles was reduced. Consequently, the number of strong pinning points was reduced, and the damping performance was enhanced. On the other hand, the aging treatment caused many second phases to precipitate, which increased the number of strong pinning points and generally weakened the damping performance of the magnesium alloy. However, some scholars [15] have found that the twin crystals produced by aging treatments exhibited smooth boundaries that were hardly pinned by the second phase. Therefore, the boundaries of twin crystals could move or slide under external loading, thereby improving the damping performance of magnesium alloys.

Although scientists have done a lot of research on ZK60 magnesium alloy [3,16,17,18,19], there are few studies on the damping behavior of high-strength ZK60 magnesium alloy prepared by large strain rolling (LSR). Therefore, in this study, high-strength ZK60 magnesium alloy sheets were prepared by LSR and subjected to an aging treatment (T5) and a solution + aging treatment (T6). The influence of different aging processes on the damping behavior of LSR-prepared ZK60 magnesium alloys was explored.

## 2. Materials and Methods

In this study, semicontinuous ZK60 ingots were used. The actual composition is shown in Table 1. First, the ingots were subjected to a homogenizing annealing treatment at 330 °C/24 h + 420 °C/4 h. After holding at 300 °C for 13 min, the specimens were subjected to LSR, followed by immediate water quenching. The LSR-prepared ZK60 magnesium alloys were then heat-treated using the processes shown in Table 2. The damping performance of the ZK60 magnesium alloys was measured with a dynamic thermomechanical analyzer (TA Instruments dynamic thermomechanical analyzer DMA 850). The experiment was performed with a strain amplitude of 0.5–1000 μm, a frequency of 1 HZ, and a sample size of 30 mm × 3 mm × 1 mm (length × width × thickness). The test mode was the single cantilever mode. During the measurement, the effective length of the specimen was 17.5 mm. The microstructure of the specimen was characterized by an optical microscope (The Leica DMI3000 M inverted, manual microscope, Guangzhou, China) and field emission scanning electron microscopy (ZEISS SIGMA 500/VP high-resolution field emission scanning electron microscope, Guangzhou, China).

## 3. Results

### 3.1. Microstructure

Figure 1 shows the microstructure of the T5- and T6-treated ZK60 magnesium alloys. The grain size of ZK60 was basically unchanged after the T5 treatment (approximately 2.9 μm). However, after solution treatment, grains grew significantly, the grain size was not uniform, recrystallization occurred, and the grain size was approximately 8.0 μm. This is because the LSR-prepared ZK60 magnesium alloy had many defects. As the holding time increased, the deformed structure disappeared and was replaced by the newly formed undistorted equiaxed crystals. As the holding time continued to increase, the recrystallized grains “consumed” each other and grew [20].

Further investigation revealed that the precipitated phases of the ZK60 magnesium alloy were mainly MgZn and MgZn_2_ (Figure 2). The amount of the second phase increased as the aging time increased (Figure 3), and the second phase was mainly distributed along the grain boundaries. This is because the recrystallized grain boundaries were high-angle grain boundaries, and the precipitated phases tended to aggregate at the grain boundaries. The granular second phase also resulted in the high total interfacial energy of the system, which increased the instability of the system. To reduce the total interfacial energy, small particles were coarsened by dissolution [21], resulting in the growth of the second phase. A comparison of the SEM images of the T5 specimen and the T6 specimen (Figure 3) showed that under the same aging conditions, the second phase of the T6 state is significantly thicker and more than the second phase of the T5 state, but the number of the second phase of the T6 state in the grains is obviously less than that of the T5 state. This is because the solution treatment makes the alloy more metastable; therefore, the second phase is easier to nucleate, and when the ZK60 magnesium alloy is treated with solution treatment, the dislocations, sub-grain boundaries and other defects generated by rolling are reduced because the effective nucleation points such as dislocations and sub-grain boundaries are reduced, resulting in the precipitation of the second phase is not easy to be dispersed [19], and easier to gather at the grain boundary. At the same time, the increase in temperature accelerated the diffusion of alloy elements, the faster the precipitation rate of the second phase, and the greater the tendency of precipitation and growth [21]. Therefore, at the same aging time, the amount of the second phase that precipitated during 190 °C aging was more than that that precipitated during 170 °C aging.

### 3.2. Damping Performance

Figure 4 shows the damping-strain amplitude curves of the T5- and T6-treated ZK60 magnesium alloys. Several findings can be drawn from Figure 1. (1) The damping performance decreased with the increase in the aging time. However, the damping performance of the 4 h-aged specimen was higher than that of the un-aged specimen. (2) At the same aging time, the damping performance of the 170 °C-aged specimen was higher than that of the 190 °C-aged specimen. At the same aging temperature, the damping performance of the T6-treated ZK60 magnesium alloy was higher than that of the T5-treated specimen. (3) The damping performance increased monotonically with the increase in strain amplitude. (4) The changes in the damping performance with strain amplitude can be clearly divided into three stages. In the first stage, the magnitude of damping was basically unchanged with the increase in strain amplitude. This stage is referred to as the strain-independent damping stage. In the second stage, when the strain exceeded the first critical strain amplitude, damping first increased slowly as the strain amplitude increased. Before the second critical point, a damping plateau appeared, in which the damping remained constant with the increase in strain amplitude. As aging progressed, the plateau gradually widened. On the other hand, the solution-treated alloy did not exhibit this plateau. In the third stage, damping increased rapidly with the increase in strain amplitude. The T5- and T6-treated specimens had different third stage critical values. The third stage critical value of the T5-treated ZK60 magnesium alloy was approximately 0.2%. It was also found that this critical value tended to increase with the increase in aging time. The third stage critical value of the T6-treated ZK60 magnesium alloy increased as the aging time increased.

The damping mechanism of magnesium and magnesium alloys is dislocation damping. According to the Granato-Lücke (G-L) model [19,22], the dislocation damping of magnesium alloys mainly consists of two types, strain-independent damping, *Q*_0_^−1^, and strain-dependent damping, *Q*_h_^−1^, so
(1)$$Q^{-1}=Q_{0}^{-1}+Q_{h}^{-1}$$

At small strain amplitude, the dislocations underwent reciprocating “bowing” movements between weak pinning points that have an average spacing of *L_C_*. The resulting consumption *Q*_0_^−1^ can be expressed as follows:(2)$$Q_{0}^{-1}=\rho BL_{C}^{4}\omega /(36Gb^{4})$$
where *ρ* is the mobile dislocation density, *G* is the shear modulus, *B* is a constant, *b* is the Burgers vector, and *ω* is the measured angular frequency.

At a large strain amplitude, the dislocations underwent reciprocating “bowing” movements between strong pinning points that have an average spacing of *L_N_*. The resulting consumption *Q*_h_^−1^ can be expressed as follows:(3)$$Q_{h}^{-1}=(C_{1}/\varepsilon_{0})\exp(-C_{2}/\varepsilon_{0})$$
(4)$$C_{1}=\Omega \rho L_{N}^{3}K\eta \alpha /(\pi^{2}L_{c}^{2})$$
(5)$$C_{2}=K\eta \alpha /L_{C}$$
where Ω is the orientation factor, *ε*_0_ is the strain amplitude, *K* is a factor related to the sample orientation and the anisotropy of the elastic coefficient, *η* is the mismatch coefficient between solute atoms and solvent atoms, and *α* is the lattice constant.

Figure 5 shows the G-L model. It can be seen that the heat treatment had a great impact on the number of and spacing between the strong and weak pinning points. When the stress was high enough, the dislocation line broke away from the weak pinning points and enhanced the damping performance [4].

From Figure 1, and Figure 4, it can be seen that during the aging process, the grain size remained unchanged, but the amount of the second phase increased as the aging time increased. The amount and size of the second phase in the 190 °C-aged specimen were significantly larger than those of the 170 °C-aged specimen. Since the second phase can form strong pinning points and can shorten *L_N_*, the damping value of the ZK60 magnesium alloy decreased with the increase in aging time in the range of 4 h–24 h. At the same aging time, the damping value of the 190 °C-treated specimens was less than that of the 170 °C-treated specimen (Figure 4).

After solution treatment, the grain size of the ZK60 magnesium alloy increased, and the number of dislocation tangles decreased substantially [23]. The reduction in these strong pinning points led to increased *L_N_* and increased dislocation mobility, thereby enhancing the damping performance. Therefore, the damping performance of the T6-treated specimen was better than that of the T5-treated specimen (Figure 4).

TEM images of the ZK60 magnesium alloy after the 400 °C + 170 °C treatment for 24 h are shown in Figure 6 shows that the number of dislocation tangles in the alloy was relatively small, so the dislocation mobility was relatively high. Therefore, the damping of the ZK60 magnesium alloy significantly increased after the solution treatment. This result is consistent with the conclusions of other studies [14,19,24].

Because of the differences between the thermal expansion coefficients of the coarse, second phase and the matrix, new dislocations will form after aging [25]. An appropriate increase in the dislocation density can improve the damping performance of the material [26]. Therefore, the damping performance of the T5- and T6-treated ZK60 magnesium alloys both increased after 4 h of aging.

There were two critical breakaway points, and a damping plateau appeared between the first and second critical breakaway points. Therefore, the damping curve obtained in this study was not a typical strain-damping spectrum. Rogers [27,28] extended the G–L dislocation damping model and believed that the occurrence of two critical breakaway points (strain amplitudes) in the damping-strain amplitude curve was due to the difference between the breakaway capabilities of the edge dislocations and screw dislocations in the alloy, i.e., screw dislocations can move more easily than edge dislocations. The G–L model only considers one type of dislocation (the edge dislocation), but in fact, the increase in the damping performance is related to the breakaway of both screw and edge dislocations. Therefore, it can be inferred that the first critical point may be the breakaway of spiral dislocations, which caused the damping value to increase slowly. However, at this time, the increase in strain amplitude was not sufficient to drive the movement of edge dislocations, so a damping plateau appeared. When the strain amplitude was high enough to drive the movement of edge dislocations, damping increased rapidly. The damping plateau became more obvious with the increase in aging time. Therefore, the distribution and amount of the second phase had a decisive influence on the damping plateau. The increase in the second phase amount made it more difficult for the edge dislocations to move.

When the strain is greater than the critical breakaway strain, *ε*_2cr_, the following equation can be obtained from Equation (3):(6)$$\ln(Q_{h}^{-1}\varepsilon )=\ln C_{1}-C_{2}/\varepsilon$$

The G–L curve of the LSR-prepared ZK60 magnesium alloy is shown in Figure 7, and the values of *C*_1_ and *C*_2_ are shown in Table 3. The G–L curves of the LSR-prepared ZK60 magnesium alloy were straight lines, indicating that the damping mechanism of the ZK60 magnesium alloy was dislocation damping. Equation (5) shows that at a certain strain amplitude, the damping value can be obtained according to the values of *C*_1_ and *C*_2_. The larger *C*_1_ is, and the smaller *C*_2_ is, the larger the *Q*_h_^−1^.

## 4. Conclusions

After the aging treatment, the grain size of the LSR-prepared ZK60 magnesium alloy did not change significantly, and the second phase increases with the extension of the aging time, resulting in a decrease in the damping performance, but when the aging time is 4 h, the damping performance is increased compared with the unaging state. After T4 treatment, the damping performance increases due to the growth of crystal grains and the reduction of dislocation entanglement.

At the same aging time, increasing the aging temperature was beneficial to the precipitation of the second phase, thereby reducing the damping performance.

Because the critical breakaway strain amplitudes of screw dislocations and edge dislocations were different, a damping plateau appeared in the damping-strain amplitude curves of the LSR-prepared ZK60 magnesium alloys, and the width of the damping platform is related to the second phase, the more the second phase score, the more diffuse the second phase and the wider the damping platform.

## Figures and Tables

**Figure 1 materials-13-05574-f001:**
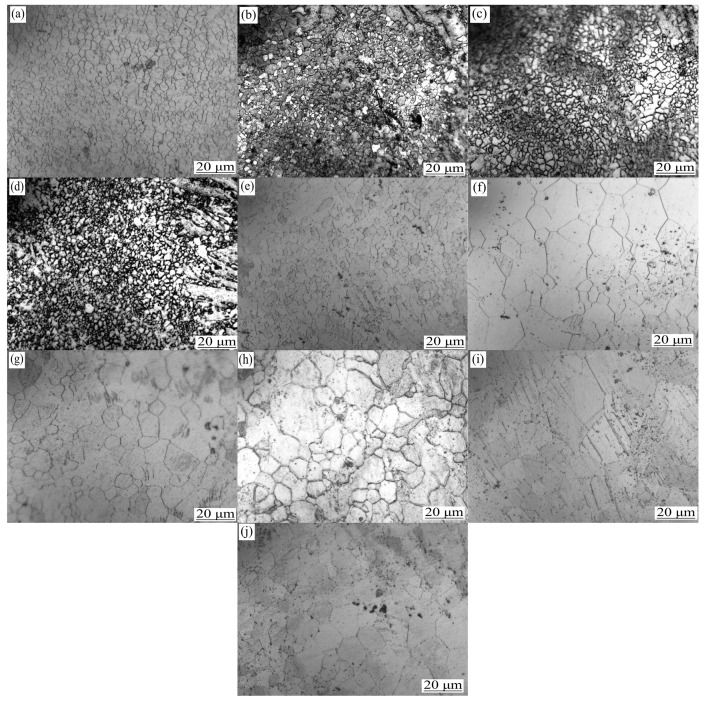
Microstructure of ZK60 magnesium in heat treat state. (**a**) Hot rolled state; (**b**) at 170 °C for 4 h; (**c**) at 170 °C for 12 h; (**d**) at 170 °C for 24 h; (**e**) at 190 °C for 12 h; (**f**) at 400 °C for 3 h; (**g**) at 400 °C for 3 h and 170 °C for 4 h; (**h**) at 400 °C for 3 h and 170 °C for 12 h; (**i**) at 400 °C for 3 h and 170 °C for 24 h; (**j**) at 400 °C for 3 h and 190 °C for 12 h.

**Figure 2 materials-13-05574-f002:**
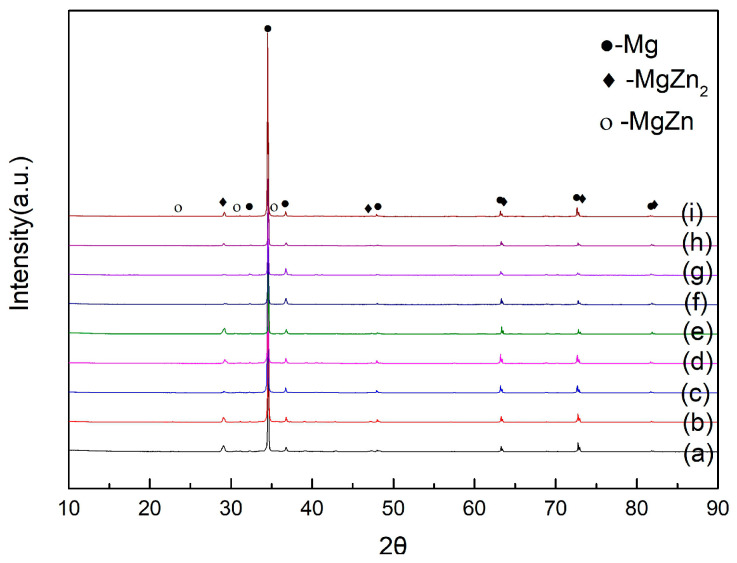
XRD diagram for different heat treatment conditions. (**a**) Hot rolled state; (**b**) at 170 °C for 4 h; (**c**) at 170 °C for 12 h; (**d**) at 170 °C for 28 h; (**e**) at 400 °C for 3 h; (**f**) at 400 °C for 3 h and 170 °C for 12 h; (**g**) at 400 °C for 3 h and 170 °C for 24 h; (**h**) at 400 °C for 3 h and 170 °C for 28 h; (**i**) at 190 °C for 12 h.

**Figure 3 materials-13-05574-f003:**
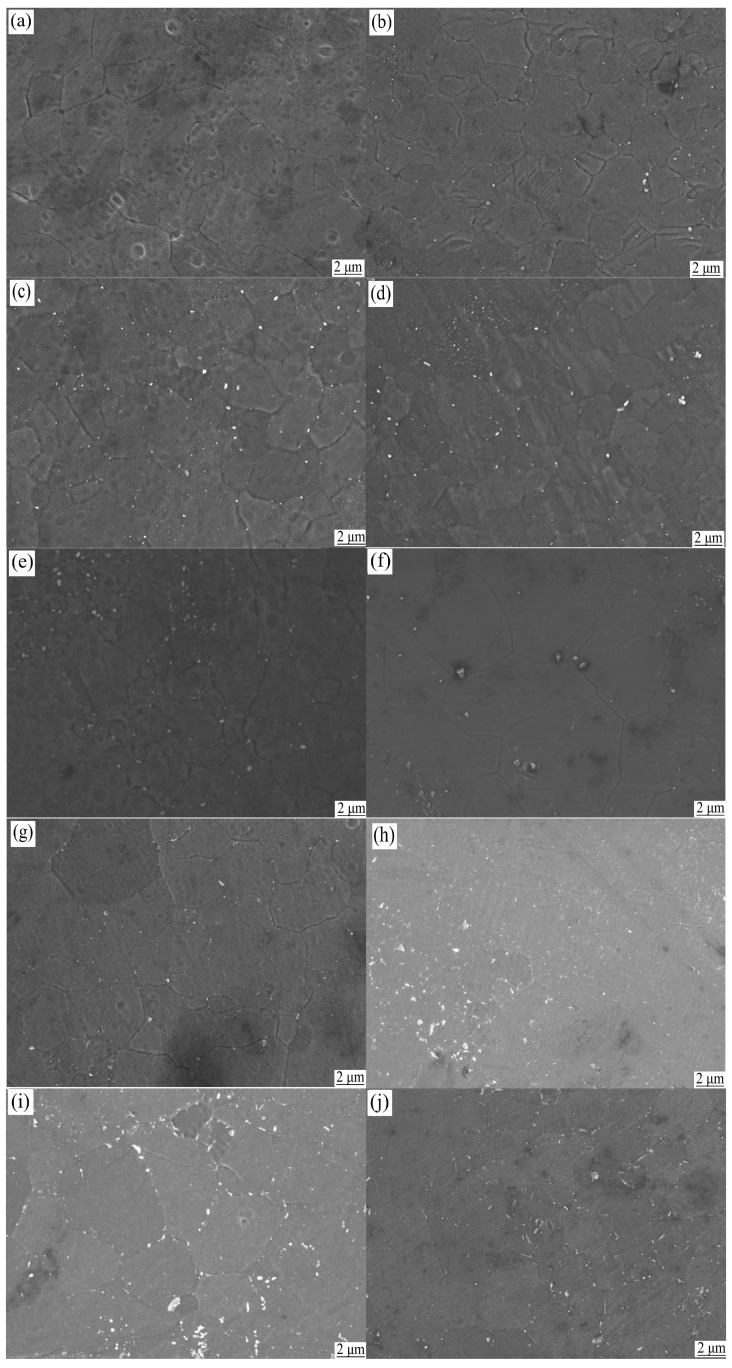
SEM of ZK60 magnesium alloy in heat treat state. (**a**) Hot rolled state; (**b**) at 170 °C for 4 h; (**c**) at 170 °C for 12 h; (**d**) at 170 °C for 24 h; (**e**) at 190 °C for 12 h; (**f**) at 400 °C for 3 h; (**g**) at 400 °C for 3 h and 170 °C for 4 h; (**h**) at 400 °C for 3 h and 170 °C for 12 h; (**i**) at 400 °C for 3 h and 170 °C for 24 h; (**j**) at 400 °C for 3 h and 190 °C for 12 h.

**Figure 4 materials-13-05574-f004:**
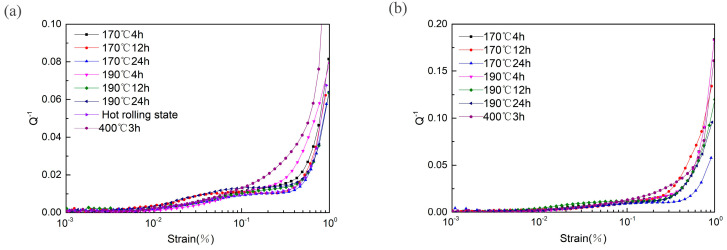
Damping-strain curve under different heat treatments. (**a**) T5 state; (**b**) T6 state.

**Figure 5 materials-13-05574-f005:**
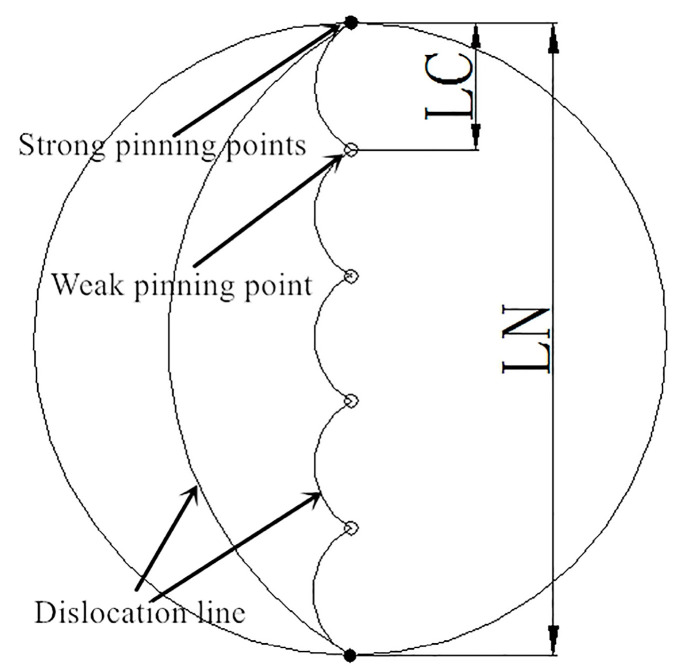
Granato-Lücke (G–L) model theory.

**Figure 6 materials-13-05574-f006:**
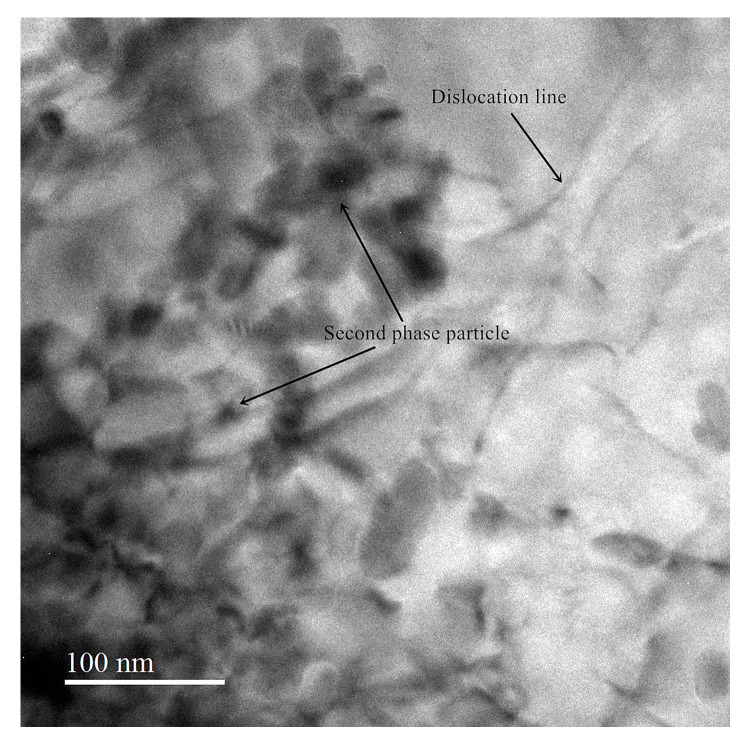
TEM image of ZK60 magnesium alloy heat-treated at 400 °C for 3 h + 170 °C for 24 h.

**Figure 7 materials-13-05574-f007:**
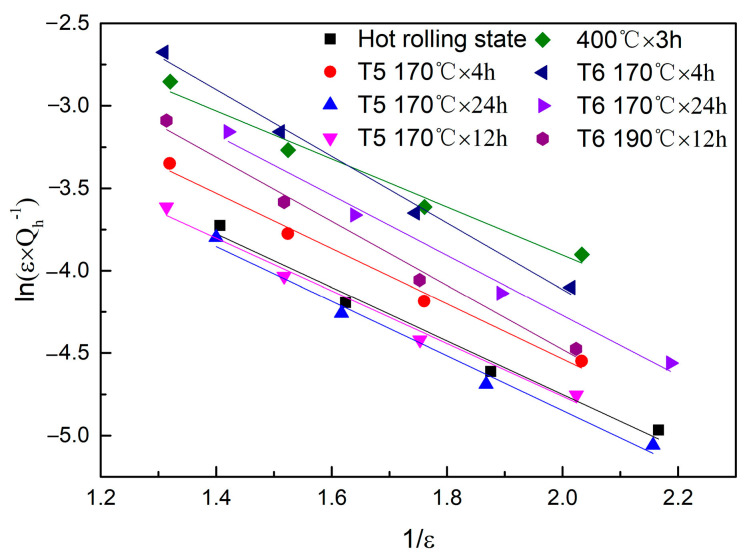
G–L curve of rolled ZK60 magnesium alloy.

**Table 1 materials-13-05574-t001:** Actual composition of ZK60 alloy (w/%).

Alloy No.	Zn	Zr	Fe	Cu	Ni	Mn	Si	Al	Mg
ZK60	5.57	0.69	0.0010	0.0032	0.0026	0.021	≤0.01	≤0.01	Bal

**Table 2 materials-13-05574-t002:** Process of heat treatment.

Process No.	Temperature	Time
T5	170 °C	4, 12, 24 h
	190 °C	4, 12, 24 h
T6	400 °C, 3 h + 170 °C	4, 12, 24 h
	400 °C, 3 h + 190 °C	4, 12, 24 h

**Table 3 materials-13-05574-t003:** Values of *C*_1_ and *C*_2_.

Heat Treatment Process	*C* _1_	*C* _2_
Hot rolling state	0.222664305	1.62422
170 °C 4 h	0.306382793	1.67621
170 °C 24 h	0.216317077	1.65848
190 °C 12 h	0.209590427	1.59952
400 °C 3 h	0.370567376	1.45567
400 °C 3 h + 170 °C 4 h	0.929117563	2.01992
400 °C 3 h + 170 °C 24 h	0.53361536	1.82115
400 °C 3 h + 190 °C 12 h	0.590261276	1.94533

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
