# Peer review of "Effects of Aging Process on the Damping Performance of ZK60 Magnesium Alloys Prepared by Large Strain Rolling"

_materials, 2020, doi:10.3390/ma13235574_

Round 1

Reviewer 1 Report

“Although scholars in China and abroad have conducted many studies on the damping mechanism of magnesium alloys, the damping mechanism of the ZK60 magnesium alloy, especially the damping mechanism of the high-strength ZK60 magnesium alloy prepared by large strain rolling (LSR), has not been thoroughly investigated.”

It's a pretty vague description because the concept of the manuscript is similar to previous studies on ZK60 using the Granato-Lücke dislocation pinning model.

For example:

[3] Xian-hua CHEN et al: Influence of impurities on damping properties of ZK60 magnesium alloy, Transactions of Nonferrous Metals Society of China, Volume 20, Issue 7, July 2010, Pages 1305-1310

"Granato and Lucke dislocation pinning model was employed to explain damping properties of the alloys"

Jing-feng WANG et al: Effects of Y on mechanical properties and damping capacity of ZK60 magnesium alloys, Transactions of Nonferrous Metals Society of China, Volume 20, Supplement 2, July 2010, Pages 366-370

"The microstructure, mechanical properties and damping capacity of ZK60-xY (x=0, 1.5%, 2.5%, 4.0%, mass fraction) magnesium alloys were investigated by using the optical microscope (OM), X-ray diffractometer (XRD), universal tensile testing machine and dynamic mechanical analyzer (DMA). The mechanisms for damping capacity of referred alloys were discussed by Granato-Lücke theory."

If the authors expect any additional damping mechanisms, they should modify the Granato-Lücke model as presented, for example, in:

Xiong-peng Zhou et al: Effects of low temperature aging precipitates on damping and mechanical properties of ZK60 magnesium alloy, Journal of Alloys and Compounds, Volume 819, 5 April 2020, 152961

"The damping capacity improvement can be explained by the self-modified Granato-Lücke model."

The present manuscript deals with damping behaviour as mentioned in the introduction:

The influence of different aging processes on the damping behavior of LSR-prepared ZK60 magnesium alloys was explored.”

The authors should emphasize the novelty of the current research and conclusions should be written on this basis. They present T5 and T6 at various conditions (temperature, time), but in the conclusions, there is no mention of whether any treatment is perspective ... Perhaps readers should get the impression that it makes sense to perform heat treatment in order to improve damping behaviour.

Furthermore, the quality of the manuscript should be improved.

For example:

optical metalloscope - optical microscope (The Leica DMI3000 M inverted, manual microscope)

the crystal grains – grains

the recrystallized grains “swallowed” - … “consumed” sounds better to me

MgZn2 - MgZn2

Q0-1 - Q0-1 ... etc.

Reference style (e.g. [3])

Fig. 6. I recommend rescaling Q-1 (e.g. 0 - 0.03) to increase visibility. Without that, it is hard to see and discuss any changes in the curves …

If the authors significantly improve the manuscript and clearly present the contribution of the results to the topic, I can recommend the manuscript for publication.

Author Response

Thanks for the suggestions given by the reviewer, I made some changes to the paper:

1.Improved the description of the introduction and added several references

2. Improved the quality of manuscripts

3.Supplements the conclusion

The main innovations are as follows: 1. Heat treatment of high-strain rolled ZK60 magnesium alloy has obtained the damping law

2.It is found that the damping performance is increased when the aging is 4h compared with the unaging time.
3. Get the influence of the second relative damping platform

The amendments to the conclusion are as follows:

  • After the aging treatment, the grain size of the LSR-prepared ZK60 magnesium alloy did not change significantly,and the second phase increases with the extension of the aging time, resulting in a decrease in the damping performance,but when the aging time is 4h, the damping performance is increased compared with the unaging state. After T4 treatment, the damping performance increases due to the growth of crystal grains and the reduction of dislocation entanglement.
  • At the same aging time, increasing the aging temperature was beneficial to the precipitation of the second phase, thereby reducing the damping performance.
  • Because the critical breakaway strain amplitudes of screw dislocations and edge dislocations were different, a damping plateau appeared in the damping-strain amplitude curves of the LSR-prepared ZK60 magnesium alloys, and the width of the damping platform is related to the second phase,the more the second phase score, the more diffuse the second phase and the wider the damping platform.

Reviewer 2 Report

Authors investigated the effect T5 and T6 heat treatment on the damping properties of ZK60 wrought magnesium alloy. The results can be of interest to readers of Materials journal.  However, before it is published certain issues are needed to be addressed.  

  • Line 73: Please provide the actual composition of the ingots used in this study. The composition provided in the Table 1 is the general composition range for this alloy.
  • Line 74: Why did authors conduct a two-stage homogenization heat treatment? Please explain the benefit of this type of treatment.
  • Line 75: Please provide the dimensions for the ingot subjected to LSR. Does the entire cross-section of the ingot heat up to 300°C in 13 minutes?
  • I recommend authors to combine Figure 1 and 2 into a single figure to provide a better comparison. Same thing should be done for Figure 4 and 5, since the size of precipitate sizes were compared for the same aging time for non-solutionized and solutionized samples.
  • In Figure 5e, the presented microstructure is for 400°C solution treatment followed by aging treatment at 190°C for 12 hours which is not comparable to Figure 4e where aging treatment at 190°C for 24 hours. Please consider presenting results for same aging times to accurate comparison.
  • What is the basis for selecting solution treatment and aging treatment conditions? When does ZK60 alloy reach to peak hardness condition for different aging temperatures? Please consider adding a paragraph describes kinetics of the precipitate formation at different temperatures and connect it to the precipitate size discussion.
  • “114 (Fig. 5) showed that under the same aging conditions, the second phase of the T6 specimen was

115 obviously coarser than that of the T5 specimen. This is because the solution treatment reduced the

116 number of defects (such as dislocations and subgrain boundaries) that were produced by LSR. As a

117 result, it is difficult for the second phase to disperse or precipitate [18], while it is easier for it to

118 aggregate at the grain boundaries. At the same time, the increase in temperature accelerated the

119 diffusion of alloy elements [17]. Therefore, at the same aging time, the amount of the second phase

120 that precipitated during 190 °C aging was more than that that precipitated during 170 °C aging.”

What is the preferred nucleation site for the MgZn and MgZn2 phases in ZK60 alloys? Solution treatment can actually accelerate the nucleation since alloy becomes more metastable and thus the driving force for nucleation is higher. Therefore, it is possible to precipitates preferentially nucleate on grain boundaries and grow rapidly. I think thorough consideration of precipitate formation and better control of microstructure is key to control damping properties. Please consider improving the section between lines 114 and 121.

Round 2

Reviewer 1 Report

The authors have improved the manuscript and I recommend it for publication. I only have two minor suggestions for editing the text.

In the abstract:

“optical metalloscope” – should be replaced by "an optical microscope"

“Although scholars at home and abroad have done a lot of research on ZK60 magnesium alloy…”

For a scientific community, it is irrelevant whether they were scientists from China (home) or abroad. I recommend replacing by:

Although scientists have done a lot of research on ZK60 magnesium alloy…

Author Response

Dear reviewer:

Thank you for your consideration and Suggestions for my paper.

I have finished revising the paper.

With best regards!

Yours sincerely

Xuhui Feng

Reviewer 2 Report

  • Author’s did not provide the measured (actual) composition of the processed and tested ingots. The composition table provided is still the nominal composition range of the ZK60 alloy.
  • I recommend Figure 1 and Figure 3 to be rearranged in the following order:

Hot Rolled State

(a)

At 400oC for 3hours

(b)

At 170oC for 4 hours

(c)

At 400oC for 3hours and 170oC for 4 hours

(d)

At 170oC for 12 hours

(d)

At 400oC for 3hours and 170oC for 12 hours

(e)

At 170oC for 24 hours

(f)

At 400oC for 3hours and 170oC for 12 hours

(g)

At 190oC for 12 hours

(h)

At 400oC for 3hours and 190oC for 12 hours

(j)

It will be easier for reader to compare the effect of heat treatment on the microstructure.

  • TEM image provided in Figure 6a is very blurry. The quality of the image must be improved.
  • The scale bars on the images are not consistent in size and length. Increase the visibility of scale bars.

Author Response

Dear reviewer:

Thank you for your consideration and Suggestions for my paper.

I made some modifications in my paper.

With best regards!

Yours best regards!

Xuhui Feng

Table1.Actual composition of ZK60 alloy (w/%)

Alloy No.

Zn

Zr

Fe

Cu

Ni

Mn

Si

Al

Mg

ZK60

5.57

0.69

0.0010

0.0032

0.0026

0.021

≤0.01

≤0.01

Bal

The name of the picture has also been changed

Round 3

Reviewer 2 Report

Author's did not address my comments reorganizing Figures 1 and 3 as I presented in my last round of review. In addition, the TEM image quality in Figure 6 is very poor and it is not publication quality which I also stated in the last review. Please address these issues. 

Author Response

Dear reviewer:

Thanks for the  suggestions,I made some improvements to paper:

  1. Adjustments have been made to Figure 1 and Figure 3.
  2. Because the morphology of the precipitated phase has little effect on the paper, Figure 6a is deleted.

With best regards!

Yours sincerely

Xuhui Feng